# Understanding the diabetes self-care behaviour in rural areas: Perspective of patients with type 2 diabetes mellitus and healthcare professionals

**Saurabh Kumar Gupta**[1], **P.V.M. Lakshmi**[1], **Venkatesan Chakrapani**[2], **Ashu Rastogi**[3], **Manmeet Kaur**[1,4]*

1 Department of Community Medicine and School of Public Health, Post-Graduate Institute of Medical Education and Research, Chandigarh, India, 2 Centre for Sexuality and Health Research and Policy (C-SHaRP), Chennai, India, 3 Department of Endocrinology, Post-Graduate Institute of Medical Education and Research, Chandigarh, India, 4 HEAL Foundation, Chandigarh, India

* mini.manmeet@gmail.com

**Data Availability Statement:** Data cannot be shared publicly because, during the informed consent process, participants did not provide

## Abstract

### Background

Diabetes self-care behaviour plays a crucial role in managing the diabetes effectively and preventing complications. Patients with type 2 diabetes mellitus (T2DM) and health care professionals (HCPs) of rural areas often face unique challenges when it comes to diabetes self-care practices (SCPs). Therefore, this study aim to explore the perspectives of patients with T2DM and HCPs on diabetes SCPs.

### Methods

Eight focus group discussions (FGDs) among individuals with T2DM and In-depth interviews (IDIs) with 15 HCPs were conducted in rural areas of Punjab, North India. Capability, Opportunity, Motivation, and Behaviour model (COM-B) was employed for thematic framework analyses.

### Results

The study participants perceived that a limited understanding of diabetes mellitus (DM), beliefs in alternative therapies, drug side effects, attitudes towards DM **(psychological capability)**, comorbidities **(physical capability),** family support **(social opportunity)**, financial and time constraints, and weather conditions **(physical opportunity)** contributed to lack of DM SCPs. Physicians' guidance and support were motivating them to adhere to SCPs, especially when aligned with their sense of self-efficacy **(reflective motivation)**. HCPs constraints in providing patient-centred care are due to training limitations **(psychological capability)** and a lack of essential resources **(physical opportunities)**. Participants expressed need for comprehensive diabetes care **(automatic motivation)** through structured diabetes education intervention to improve diabetes SCPs.

permission for the public use of their data. The authors will make the anonymized data sets available for other researchers upon request, after receiving a permission from the institutional ethics committee of PGIMER, Chandigarh. Data requests can be directed to the corresponding author of the manuscript (email: mini.manmeet@gmail.com) or to the ethics committee directly (email: iecpgi@gmail.com).

**Funding:** The first (SKG) author was a recipient of Indian Council of Medical Research – JRF/SRF Fellowship Scheme [No. 3/1/3/JRF-2016/HRD-90)] for pursuing his Ph.D.. However, The ICMR-JRF/SRF scheme had no role in study design, data collection and analysis, decision to publish, or preparation of the manuscript. The authors did not receive any specific funding for this work.

**Competing interests:** The authors have declared that no competing interests exist.

## Conclusions

The study findings indicate that various factors influence diabetes SCPs from the perspectives of both patients with T2DM and HCPs and emphasizes the need for a multi-faceted approach to improve diabetes SCPs in rural areas. Implementing a structured diabetes self-care intervention strategy in rural areas may help for preventing and mitigating the impact of diabetes-related complications in rural areas.

## 1. Introduction

Diabetes SCPs are essential in diabetes mellitus (DM) treatment for achieving the optimal glycemic target [1, 2]. Diabetes SCPs include healthy eating habits, physical activity, medication adherence, monitoring blood glucose as prescribed, regular follow-ups, foot care and healthy coping [3]. Adults with T2DM who live in resource-constrained health settings face a variety of barriers to SCPs, including restricted access to quality healthcare (lack of medicines, lack of physicians, lack of blood glucose monitoring facilities), cultural (inappropriate dietary behaviours, faith in herbal medication, myths about DM) and sociopsychological issues (lack of knowledge and skills, lack of family supports, diabetes-related distress) that have significant social and economic implications [4, 5]. Numerous quantitative cross-sectional studies have been conducted in South Asian countries to evaluate DM SCPs, and the findings indicate that patients with T2DM have suboptimal DM SCPs [6]. Quantitative studies use closed-ended structured questions with limited response options and cannot capture the depth and complexity of the factors influencing self-care behaviours, while qualitative studies provide better insight using key stakeholder perspectives [7]. Thus, using a qualitative research approach enables the development of strategies appropriate to help human behaviour change.

Quantitative studies and many qualitative studies may identify barriers and opportunities related to diabetes self-care practices, but they often do not provide an in-depth understanding of the underlying context in which these factors occur [8]. Theories and models, on the other hand, can help make connections and provide a framework for understanding and analyzing the complex factors influencing behavior change and SCPs [9].

Models like the Health Belief Model (HBM) and the Transtheoretical Model (TTM) have been widely used in health behavior research, including in the context of diabetes SCP [10]. The HBM focuses on individuals' perceptions of health threats, benefits of action, and barriers to change, while the Transtheoretical Model describes the stages individuals go through when making behavior changes.

Holistic models such as the Theoretical Domains Framework (TDF) and the Behavior Change Wheel (BCW) have gained prominence in recent years [11]. These models incorporate multiple theories and provide a comprehensive approach to understanding behavior change. The TDF integrates various psychological theories and identifies a range of factors influencing behavior change, including knowledge, beliefs, social influences, and environmental factors. The BCW, in turn, provides a systematic approach to designing interventions by aligning behavior change techniques with the specific context and target behavior [12].

The COM-B has been successfully used in many intervention studies e.g., healthy eating habits, physical activity, smoking cessation, type-1 diabetes self-care; gestational diabetes mellitus; medication adherence, and HIV prevention [13, 14]. It has rarely been used for better understanding of patients' with T2DM and the HCPs' perspectives on diabetes self-care behaviour [15]. We did not find any study that explored the perspectives of patients with T2DM and

primary health care professionals of rural resource-constrained health settings on DM self-care behaviour based on the theory of behaviour change in India. Therefore, this study was undertaken to contextualise the patients with T2DM and HCPs' perspectives on diabetes SCPs using the COM-B model in a rural health setting of Punjab, North India.

## 2. Materials and methods

### 2.1 Study design and setting

A qualitative descriptive-interpretative study was done in the purposively selected district Fatehgarh Sahib of Punjab, North India, from April 2019 to January 2020 (**Fig 1**) [16]. The Fatehgarh Sahib district of Punjab was purposively selected as it is showing an increasing prevalence of T2DM in Punjab [17]. Furthermore, this district falls in the field practice area of the Department of Community Medicine and School of Public Health, PGIMER, Chandigarh. Thus, we already had good rapport with the local communities, a desirable feature in a qualitative study for good quality of data.

Manuscript was reported based on the Consolidated Criteria for Reporting Qualitative Reporting Qualitative study checklist (**S1 Appendix**).

### 2.2 Participants, sampling and recruitments

Participants of FGDs and IDIs were selected using a purposive sampling technique. FGD participants were pair matched utilising the concept of maximum variation sampling by sex, age, and socioeconomic status(S2 Appendix).

i. **Selection of FGD participants**: 1) patient with T2DM with a known medical history and on oral hypoglycemic agents (OHAs) for more than six months, 2) age $\geq$ 30 years, and 3) willing to participate in the study. Patients with type-1 diabetes mellitus and additional co-morbid conditions such as acute heart failure or acute coronary syndrome, end-stage kidney disease, cognitive impairment, pregnant women, and people unable to interact or speak were excluded from the study.

ii. **Selection of IDI participants**: MOs and ANMs who were directly involved in the management (treatment, follow-ups, and patient education) of patients with T2DM in a rural health setting for the last year were included if they were willing to participate in the study.

### 2.3 Data collection

FGDs were conducted at preferred venues (e.g., Gurudwaras [religious places], community or centres) decided by the study participants, and IDIs were carried out at the rural health facilities. Before each FGD, participants were provided (those who could read) and explained (those who could not read) the participant information sheet in the vernacular language (Punjabi). Those interested in participating signed a consent form allowing us to audio-record FGDs and IDIs. A researcher (male) with a Master's in Public Health (MPH) degree moderated each FGD and IDI, he had more than four years of experience in the development and implementation of non-communicable disease related interventions, and he was assisted by another researcher (female) with an MSc in Nursing qualification as a notetaker who had more than 12 years of experience in field of rural community-based health promotion activities. Moreover, before the commencement of the study, the two researchers had received training from experts in qualitative research who had more than 30 years of experience in field of health promotion and qualitative research methodology. Interviewer's and notetaker's

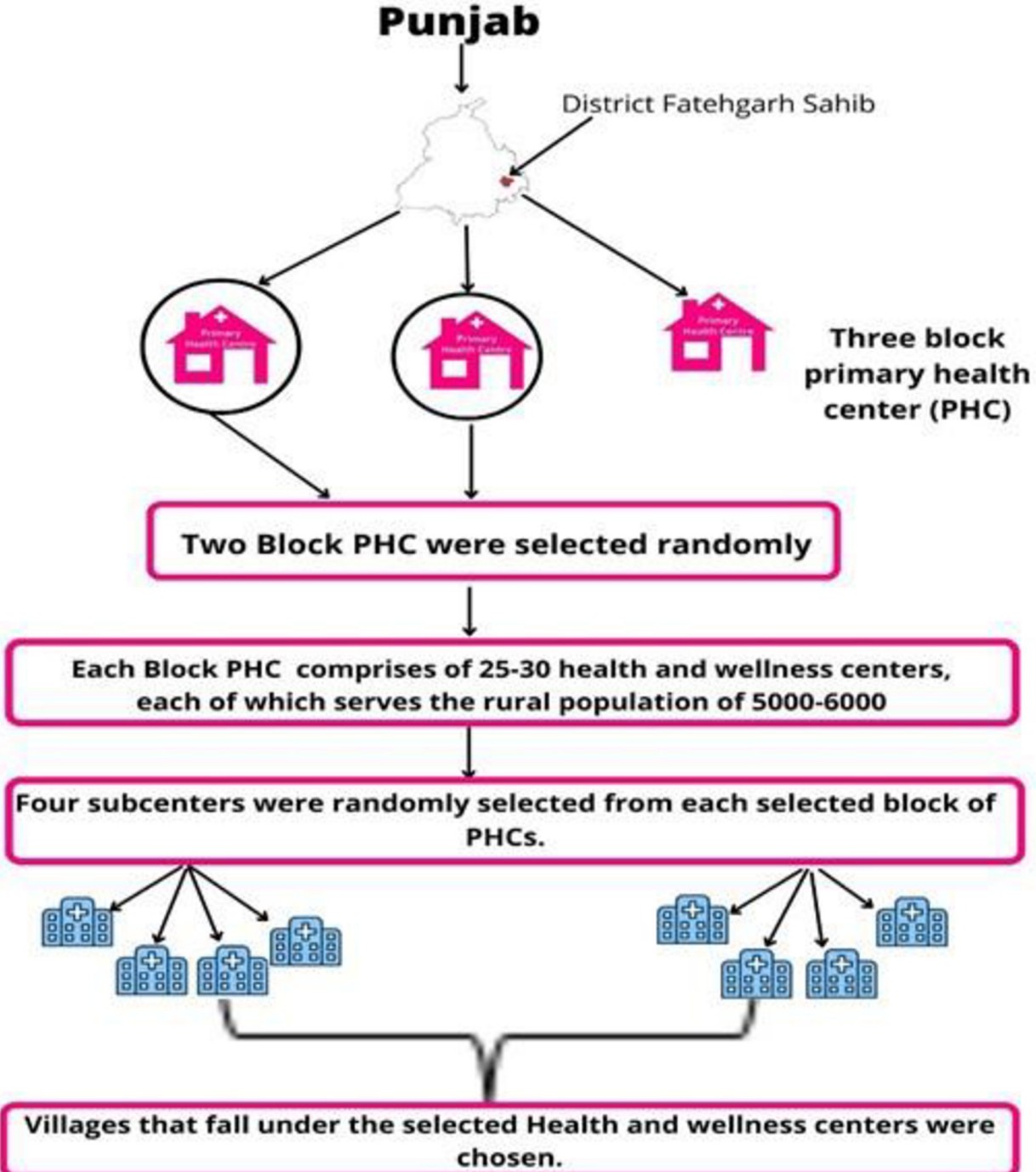

**Fig 1. Flow diagram of selection of study area.**

background allowed us in combining medical insights with sociological perspectives and thus enriching the study, and also offering a more holistic understanding of diabetes self-management within rural contexts with a focus on social structures, cultural influences, and community dynamics [18]. For each FGD and IDI, a topic guide aligned with the research objective was used to navigate the discussions and interviews (**S3 Appendix**).

Where necessary, the note taker assisted the moderator using appropriate probes to encourage more details or clarity in the participants' responses. The duration of FGDs and IDIs ranged between 30 and 45 minutes. At the end of each FGD and IDI session, participants were asked whether they wished to modify or add any information after the moderator summarised the main points of the discussions. Eight focus group discussions comprised 54 individuals with T2DM, whereas IDIs included 15 HCPs' (6 MOs and 9 ANMs). Each FGD had 6–8 participants. FGDs and IDIs were recorded and transcribed verbatim for analysis.

## 2.4 Data analysis

Using iterative process, the data collection and analysis took place simultaneously [19]. FGDs and IDIs data were processed using NVivo 11 software. Data were analysed using a thematic framework analysis approach [20]. Within each FGD and IDI, the authors focused on key terms and repeating phrases to generate codes (emergent codes) and improved the 'apriori codes'.

A qualitative research expert from the research team (Female) who had experience in the qualitative research methodology also involved in the monitoring of the FGD and IDI throughout the study period. She cross checked transcripts to enhance the credibility (researcher triangulation) and transferability of the coding and themes. Reflective memos were used to augment the coding framework. The research team frequently discussed maintaining uniformity in the codes and categories development.

The COM-B model was used for thematic framework analysis to understand the diabetes self-care practices which included medication adherence, dietary changes, physical activity, glucose monitoring, foot care, stress management or follow-ups suggested by the treating physician. The capability assessment was based on the understanding, knowledge, skills, and physical or psychological ability related to diabetes self-care practices specifically understanding of diabetes management, self-care skills, and any physical or cognitive limitations they may have. Opportunities were evaluated based on the environmental, social, and cultural factors that influence the individual's opportunities to engage in diabetes self-care practices which were healthcare services, availability of resources, family or social support, and other contextual factors. For individual's motivation to engage in the behaviour, both reflective and automatic the role of social and psychological factors such as social norms, incentives, and emotional responses were considered.

To support methodological rigour during analysis, the research team members having diverse professional backgrounds (in endocrinology, public health, and health promotion) contributed to the discussions and offered various points of view (interdisciplinary and researcher triangulation).

## 3. Results

### 3.1 Socio-demographic characteristics of the FGD participants

Each FGD comprised 6–8 participants. Most participants were male (n = 30;56%) and in the 30–60 years of age group range (n = 30;50%). The majority, (n = 46;85%), had T2DM for less than ten years (Table 1).Most of the participants were either farmers (41%) or homemakers (28%). They were almost equally distributed across all income groups. Half of the participants were illiterate.

**Table 1. Socio-demographic characteristics of FGD participants (N = 54).**

| Variables | n (%) |
|---|---|
| **Age (In yrs)** | |
| 30–59 | 30(56%) |
| ≥60 | 24(44%) |
| **Gender** | |
| Female | 24(44%) |
| Male | 30(56%) |
| **Education** | |
| Illiterate | 27(50%) |
| Literate | 27(50%) |
| **Occupation** | |
| Business/Shop | 2(4%) |
| Farmer | 22(41%) |
| Govt. employee | 2(4%) |
| Home Maker | 15(28%) |
| Labourer | 13(24%) |
| **Duration of diabetes (in yrs)** | |
| ≤10 | 46(85%) |
| >10 | 8(15%) |
| **Monthly income (INR)** | |
| <39999 | 22(41%) |
| 39999–59999 | 13(24%) |
| >59999 | 19(35%) |

**Abbreviation:** INR-Indian rupee; Yrs- Years; FGD- Focus group discussions

## 3.2 Socio-demographic characteristics of the IDI participants

The median (IQR) age of the MOs was 38 (35–41) years, whereas the mean (±SD) age of the ANM was 36 (±5) years. The average experience of ANMs was 8 years (±3years), compared to medical officers' median experience of 8 years (IQR: 6–9 years).

## 3.3 Findings

Findings were contextualised using the constructs of the COM-B model (**Fig 2**).

Participants' sample statements (quotes) are presented to support the identified codes. Participants' quotes were labelled based on focus group (FG) number, gender (Male-M, Female-F), age (<60 years for 30–60 and ≥60 years), education level (Literate-L and Illiterate -IL), and income group {Low Income Group (LIG)-≤40K, Middle Income Group (MIG)- 40K-60K, Higher Income Group (HIG) - ≥60K}. While IDI participants quotes are presented as ANM for Auxiliary Nursing Midwife IDI and MO for medical officer IDI.

### 3.3.1 Capability

**. a) Psychological capability**

Participants were asked about their perceptions of DM (causes, complications and management) to know their knowledge and awareness of DM. The findings revealed that although the participants were familiar with some aspects of DM, they were not fully informed of the crucial components of DM.

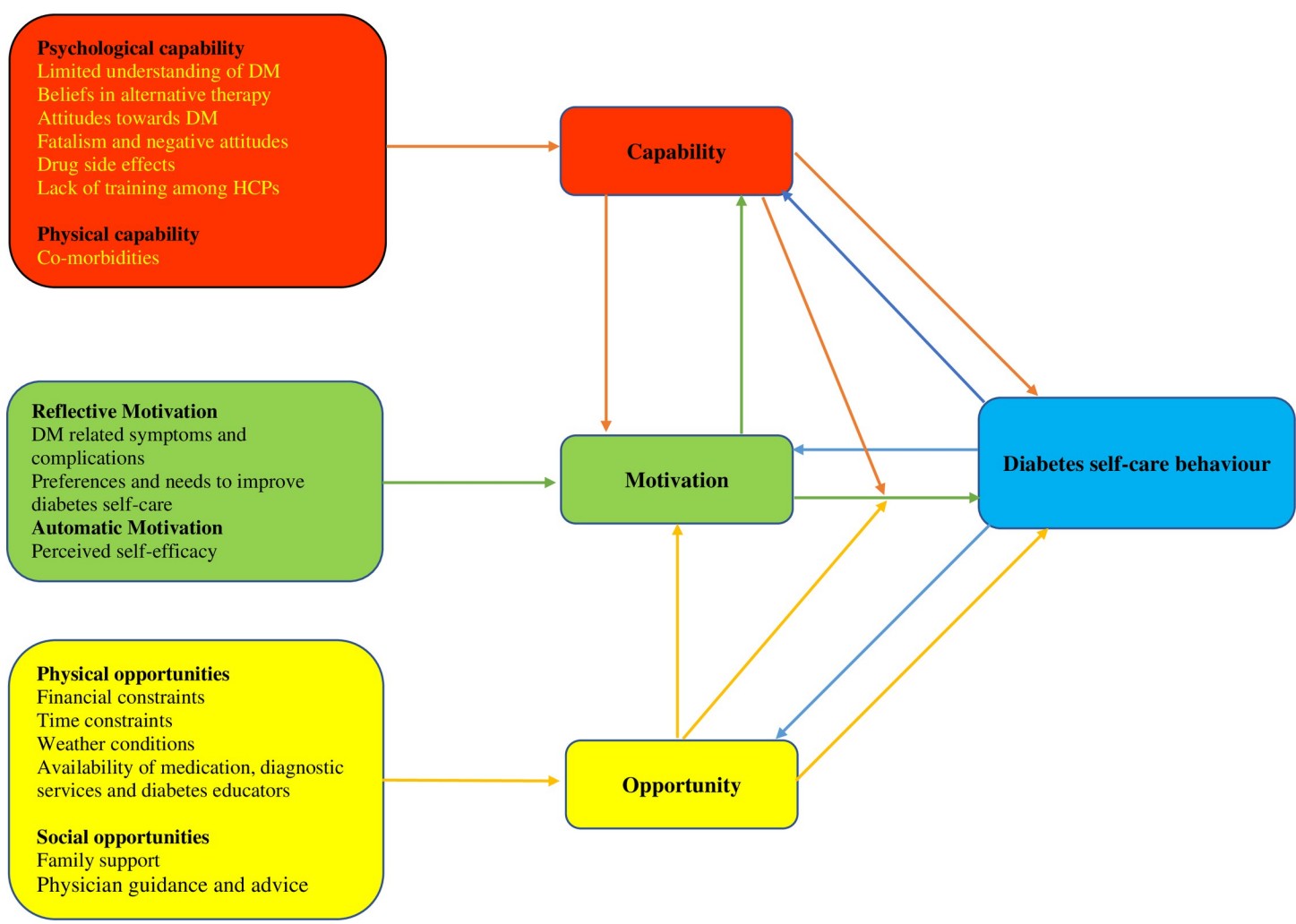

**Fig 2. Factors influencing diabetes self-care behaviour among people living with T2DM: Applying the COM-B model.**

### i. Perspective on DM

Knowledge gaps were observed as perceived barriers to optimal diabetes self-care. The majority of diabetes patients, according to the discussion, believed that consuming too much table sugar or foods sweetened with sugar was the cause of their disease. They also thought substituting jaggery powder for table sugar would not harm their health.

*'Diabetes is caused by overeating chini (table sugar) . . .. when the body cannot digest sugar-sweetened food.'* (FG2, M, >60yrs, L, MIG)Some participants discussed and believed that supernatural forces caused diabetes and that once you had it, you could do nothing to reverse it. e.g.

*'Diabetes is a curse of God. . .. . . once it has been diagnosed, there is nothing you can do to cure it.'* (FG4, F, <60yrs, L, LIG)

However, those who interacted with doctors/health care providers were well informed about general symptoms of T2DM, e.g.,

'*A doctor once told me that some of the symptoms of T2DM include. . . . . . thirst, an increase in appetite, frequent urination, and laziness.*' (FG3, M, <60yrs, IL, LIG).

The ANM and MO stated they lacked the necessary knowledge and training about diabetes self-care components.

'*I have not received any further training to teach, counsel, or encourage diabetes self-care practices among patients with T2DM, except the training I received during my studies.*' (ANM2)

'*I need proper and further training in managing diabetes. I try to apply the concepts covered in the textbooks directly.*' (MO5)

### ii. Perspective on diabetes risk factors

Participants had limited information about risk factors of T2DM development; as a result, they only perceived that having diabetes in the family is the sole cause of diabetes.

'*If a family member already has diabetes. . . .., you will surely develop it as well (diabetes), and it cannot be prevented.*' (FG8, F, >60yrs, L, HIG)ANMs also acknowledged that they lack an understanding of DM risk factors.

'*I don't know the exact causes of diabetes, but I believe it to be a hereditary condition rather than a communicable disease. I don't know what normal fasting blood sugar is? . . . But I think normal random blood sugar is approx. 150 mg/dl.*' (ANM5)

### iii. Beliefs on Alternative therapy

Participants perceived that herbal remedies might cure diabetes. This misunderstanding was largely spread by patients' relatives, family members, and advertisements. This false belief frequently made it more difficult for patients to follow instructions for diabetes self-care.

'*One of my relatives told me that herbal medicines can cure diabetes. . . . . . . . .since then, I have been taking herbal medicine.*' (FG8, F, >60yrs, L, HIG)FGD participants also misperceived multiple daily dosages of allopathy medication; thus, they skipped their medicines and took herbal medicine they thought could cure all the diseases.

'*I take one herbal medicine daily. . . . . ...which can cure multiple diseases . . ..it is very difficult to take many allopathy medicines many times a day.* (FG6, M, > 60yrs IL, MIG)

Many medical professionals expressed their concern regarding patients who use alternative medicines.

'*Because there are so many misconceptions regarding diabetes and its treatment, almost all patients in rural areas choose to visit traditional healers rather than doctors.*' (MO1)

### iv. Concern about medicine side effects

Participants perceived that the side effects of medicine prevent them from taking oral hypo-glycemic medication regularly. As a result, they skip their medication or are not taking it as prescribed.

'I don't want *to take allopathy medicine regularly. . . . . . . . . .it caused me acidity.'* (FG3, M, ≥60yrs, L, MIG)

*'I do not take medicine regularly. . . . . . . . .. because once you take it regularly, you have to take lifelong, which may be dangerous for my body. . . . . . . . ..so I skip my sugar and BP medicine for 2–3 days in a week.'* (FG8, F, <60yrs, L, HIG)

IDI participants stated that a key reason for the non-adherence to medication among patients with T2DM was their perception of its side effects.

*'Due to their misunderstanding of the side effects of diabetes medications and their lack of knowledge on the progression and management of their condition, most T2DM patients stop taking their medications.' (MO6)*

### v. Attitudes towards DM

Participants expressed negative perceptions toward self-care in general and repeatedly stated that they would die whether or not they followed their diabetes self-care routines.

*'I don't do anything to control my diabetes. . . ...drink tea with table sugar or jaggery powder and consume all sugar-sweetened foods because . . . . . .I believe that no matter what I do, I will die.'* (FG3, M, >60yrs, IL, MIG)

IDI participants stated that patients have a very casual approach towards diabetes. They ignore diabetes because of their negative attitude towards life.

*'One of the patients who came to our health centre yesterday said I have to die one day, so I will not take medicine for diabetes.' (MO4)*

*'Some diabetes patients think that DM will be cured; after receiving treatment for a while, they stop their follow-up.' (MO5)*

### vi. Perceived challenges in DM SCPs

Perceptions of routine were elicited by discussing their difficulties in their daily routines while making lifestyle and behaviour changes for optimal glycemic levels. Participants expressed how they become accustomed to eating some dishes almost constantly. As a result, many found it difficult to follow the advice that told them to either completely avoid such foods or cut back on their consumption.

*'Without ghee/dalda (clarified butter/hydrogenated vegetable cooking oil), I can't eat fulke (Indian bread).'* (FG5, M, <60yrs, IL, HIG)

Most often cited challenges to regular physical activity, such as lack of motivation, will-power, and not having created the habit of exercising. Some participants also discussed that they misperceived that doing home chores are sufficient for the body to control blood sugar.

*'I don't do any exercise…….I think doing home chores are sufficient for my body to control its blood sugar level. I don't go for a walk; I always prefer being at home.'* (FG 3, M, <60yrs, IL, LIG)

Participants believed they lacked adequate knowledge of foot care, blood sugar control, and methods to reduce diabetes care stress.

*'I have no idea how to care for my feet … Nobody (the healthcare professionals) informed me. When dealing with managing my diabetes and other issues (both at home and at work), I sometimes feel very frustrated.'* (FG5, M, <60yrs, IL, HIG)

Participants in the IDI cited the lack of knowledge and awareness about the appropriate kind and level of physical activities appropriate as the cause of inadequate of physical activities among T2DM patients.

*'The majority of patients from rural areas are uneducated. They cannot comprehend medication dosage recommendations, especially about a healthy diet and physical activity. Even after receiving counselling, individuals still do not know how often each day to take their diabetes medications and type of diets.'* (ANM 8)

#### b) Physical capability

##### i. Comorbidities

Participants discussed that their physical condition or comorbidities (e.g., high BP, cardio-vascular diseases, and knee pains) prevented them from doing regular physical activity.

*'Not only do I have diabetes, …….. but I also suffer from high blood pressure and cardiac issues…..All of this prevents me from really participating in the exercise.'* (FG3, M, >60yrs, L, MIG)

Many IDI participants also mentioned how patients' other medical or physical problems prevented them from adhering to their regular diabetes care.

*'Most T2DM patients also struggle with additional conditions like hypertension, thyroid problems, and chronic obstructive pulmonary disease, making it difficult to follow their regular diabetes care schedules like physical activity.'* (MO6)

#### 3.3.2 Opportunity.

### a) Social opportunity

#### i. Family support

Participants in the FGD frequently stated that their families did not support them enough, mainly when preparing appropriate meals for people with diabetes or reminding them to take their medications timely.

*'Whatever food is prepared in my home, I have to eat those only;. . . . . . it is very difficult for us to take care of our diet.'* (FG4, F, <60yrs, IL, MIG)

Participants in the IDI often stated that poor adherence to medication and healthy eating habits was caused by family members' lack of understanding about diabetes.

*'Family members of patients commonly don't care about medication continuation in rural areas, and they don't stop them from consuming unhealthy diets. Instead, they serve whatever food prepare for other family members, such as tea with lots of table sugar or jaggery powder and food with lots of oil and salts.'* (ANM6)

### ii. Physicians advise & support

Physician support and advice were perceived facilitators for routine diabetes care in all the FGDs.

*'My blood sugar was out of control six months before . . . It was around 350–400 mg/dl after taking food. I take medicines prescribed by my doctor.. . .it drops to around 175mg/d (random plasma blood glucose).'* (FG4, F, <60yrs, L, LIG)

### b) Physical opportunity

#### i. Financial constraints

Participants often stated that their financial situation acted as a barrier due to costs associated with diabetic self-care, especially When patients had to purchase medications that weren't available from public health facilities.

*'Now a day's medicine which I am taking is very costly, so I cannot buy it from outside (chemist shop).'* (FG4, F, <60yrs, L, LIG)

Diet, follow-ups and SMBG were other self-care practices impacted by this barrier.

*'I don't have access to a blood sugar checking facility in my village dispensary. . . .and it is also costly when I have to go a private lab. . . . . .. which I cannot afford.'* (FG6, > 60yrs, M, L, MIG)

IDI participants cited that the inadequacy of medicine or supply in the health care system makes it difficult to prescribe effective diabetes medications because many patients cannot afford them.

*'Most of the time, there are no medicines in stock, and patients cannot afford to purchase them regularly from pharmacists. Because we lack a diagnostic testing and follow-up facility,*

*patients are often referred to a secondary or tertiary care hospital. But the transportation expense and the long wait times discourage patients from visiting.'* (MO6)

### ii. Time constraints

Many participants perceived it was difficult to adhere to self-care activities, namely exercise and diet, because of their job or house-related responsibilities.

*'I usually forget to take medicine, Until I don't experience any symptoms or lack time due to busy work schedules. . .sometime also for the follow-ups as suggested by doctors.'* (FG2, M, >60yrs, L, MIG)

Most medical officers stated that they typically don't have enough time to counsel patients appropriately and would also like to help other medical staff guide patients with T2DM to handle the patient load in OPDs successfully.

*'The absence of a committed caregiver, specially trained in diabetes management, is important since they can make a substantial contribution because we often do not have enough time to counsel the patients.'* (MO 4)

### iii. Weather condition

According to some respondents, the adverse weather conditions in the winter and summer make it challenging to engage in regular physical activity.

*'I usually do a morning walk for 25 to 30 minutes daily, except when the weather is unfavourable, such as rain, heat, or cold.'* (FG1, F, <60yrs, IL, HIG)

### 3.3.3 Motivation
### . a) Reflective motivation

### i. DM-related symptoms and complications

Participants discussed that fear of symptoms and complications are facilitators for some self-care practices, especially adherence to medications and a healthy diet.

*'I never consider stopping my therapy since my blood sugar levels would rise and we would experience physical symptoms again. . . . . . like frequent urination if I stop taking my medicine.'* (FG2, M, >60y, L, MIG)

### ii. Preferences and needs to improve diabetes self-care

Participants believed that having an education programme would help them acquire self-discipline, gain new knowledge from a reliable source, and increase motivation for optimal diabetes self-care. The FGD participants tended to believe that the problem is a lack of correct information, guidance, and application to do diabetes care regularly:

All participants agreed that including demonstrations of different self-care practises, such as blood glucose monitoring and foot care, in clinical routine would help patients learn how to do them at home and develop these habits, making them less likely to forget.

*'Demonstration of blood glucose monitoring and foot care . . . . . .involvement of family members would benefit our diabetes self-care learnings."* (FG5, M, <60yrs, IL, HIG) All FGD participants perceived pamphlets, group education, and individual counselling as a mode of self-care education.

*'Educational materials like pamphlets, as well as counselling, will benefit our diabetes self-care learnings.'* (FG8, F, >60yrs, L, HIG)

The participants saw the local health professional as the best to offer the education.

*'A medical officer or community health worker (ANM or MPHW) would be the ideal person to provide the education, which should be provided during clinic visits.'* (FG7, M, <60yrs, L, LIG)

Participants identified family members as extremely important in supporting DM SCPs and counselling.

*'My husband is very supportive, but I also want our children to participate in the training because they are the ones who prepare our meals.'* (FG3, F, <60yrs, IL, LIG)

Most IDI participants believed and said that task sharing among other available healthcare professionals will improve the treatment of diabetes or any other chronic condition. Also, they emphasised the value of holistic treatment for diabetes management, including the involvement of nutritionists, counsellors, and specialists in diabetes management.

*'A trained multipurpose health care workers can significantly contribute to diabetes self-care practises.'* (MO6)

*'A multidisciplinary approach is necessary for comprehensive diabetes management.'* (ANM 8)

### b) Automatic motivation

#### i. Perceived self-efficacy

Participants perceived self-efficacy in changing their dietary habits and sedentary lifestyle and also as a facilitator for diabetes self-care.

*'I can try to take tea without sugar, go for a regular walk and reduce sugar intake.'* (FG8, F, >60yrs, L, HIG)

## 4. Discussion

Our philosophical stance, rooted in constructivism and pragmatism, acknowledges the subjective nature of knowledge formation, considering individual experiences, beliefs, and societal

contexts. This perspective allows us to understand the diverse viewpoints of both patients and healthcare providers, particularly regarding diabetes self-care practices in rural areas [21]. By embracing pragmatism, we aim not only to grasp lived experiences but also to derive actionable insights for enhancing diabetes self-care practices among people with T2DM in rural areas [22]. Balancing qualitative interpretation with a focus on practical application, our approach seeks to comprehensively explore the complexities of T2DM self-care behaviours in rural health care settings. The discussion focusses on each of the component of COM-B model.

### 4.1 Capability

The participants experienced difficulties in their ability to follow proper DMSCPs due to treatment-related myths. These were not different from the existing literature [15]. The misunderstandings regarding the cause of diabetes and its treatment identified in this study as barriers to DMSCPs differ from those discovered in research from high-income countries, that often highlights lack of awareness of diabetes and its treatment as a barrier to DMSCPs [23].

Inaccuracy and lack of knowledge could be attributable to a lack of specific and comprehensive education and poor comprehension due to low literacy levels in rural areas was clearly expressed individually during IDIs. However, it's not always the case that having formal education leads to better DMSCPs while having an in-depth understanding of the illness, its progression and the interplay of various factors is also important [24]. A link between DM awareness, DMSCPs, and glycaemic control has already been established [25] as has been in indicated in this study as well.

Some components of diabetes SCPs e.g., a healthy diet, exercise, and medication adherence were expressed clearly by most participants, but none of the participant knew about foot care, stress management, and regular MBG. This may be because their treating physicians are not emphasising on these self-care practices enough, as reported in earlier qualitative studies [26, 27]. As in most other studies, only few individuals performed foot care, and most had severe diabetes distress [28, 29].

IDI participants perceived negative attitudes and hopeless behaviour impacted adherence to DMSCPs among patients with T2DM due to a lack of knowledge and seriousness about DM, denial, or refusal to accept a DM diagnosis. The findings highlight a critical issue surrounding negative perceptions toward diabetes self-care, which might significantly impact people with DM adherence to self-care practices. This aligns with previous research emphasizing the influence of nihilistic attitude on health behaviours and chronic disease management [30]. In the current study the negative attitude toward diabetes self-care may be due to multifaceted factors such as cultural beliefs, socioeconomic challenges, availability of limited and equitable health care resources and psychological distress (diabetes related distress) [31]. Cultural influences often shape individuals' perceptions of illness and death, impacting their approach to managing chronic conditions [32]. The perception of negative attitudes, particularly in older adults with type 2 diabetes, might originate from co-morbidities and various illnesses, as well as a certainty of end-stage complications [33]. The fear of succumbing to complications of diabetes pervades individuals with T2DM due to the witnessed impact on family and community [34, 35]. Additionally, socioeconomic barriers like limited access to healthcare resources or financial constraints might lead to hopelessness, contributing to the belief that self-care efforts are futile [8]. Moreover, the psychological burden of living with a chronic condition (multimorbid conditions), inadequate support systems, could foster a fatalistic mindset, impacting people with T2DM outlook on the effectiveness of self-care practices [29, 36]. These interconnected factors might collectively shape the nihilistic perspective toward diabetes management.

FGD participants' perceptions of the side effects of continuous intake of medicines, misconceptions about the cure of diabetes and beliefs on alternate therapy for diabetes treatment support this perspective. Addressing these perspectives are essential, as they contribute to non-adherence from DM self-care practices, echoing the necessity for tailored interventions to reshape patient beliefs and attitudes toward chronic disease management [30, 37].

Furthermore, participants in the FGD stated that they faced difficulties adhering to their regular diabetes care routine. Specifically, they struggled to maintain a healthy diet, exercise regularly, and take their medications as prescribed. The most significant barriers to following healthy eating habits and medication adherence highlighted by the participants were financial constraints and family support. According to IDI participants, family support encouraged and facilitated SCPs of T2DM patients. Unhealthy dietary behaviour may also attribute to the cultural and social environment. Our results are consistent with the previous study [38]. HCPs may suggest culturally oriented, gradual alterations instead of sudden, significant ones to encourage individuals with T2DM to make dietary changes; This approach will make adjusting to a new diet less intimidating, more realistic, and easier to handle. A study in North India showed that a culturally tailored and group-based intervention strategy could increase fruit and vegetable consumption [39].

Engaging in regular physical activity, like going to the gym, may not be realistic due to financial restrictions and other issues. The lack of physical activity among the participants may also be attributed to laziness, lack of knowledge about proper workouts, and a lack of enthusiasm. The level of motivation a patient possesses can greatly impact the effectiveness of their self-care routine. Therefore, healthcare professionals may enhance patient motivation and commitment to DMSCPs by educating them about the importance of regular exercise and the benefits of doing so, as well as by encouraging them to adopt exercise routines that are simple and affordable to practise both at home and at work. HCPs in IDI expressed that they do not have enough training to provide knowledge and demonstration of DM SCPs. Although HCPs have the desire to empower their patients to manage their diabetes independently, they lack the necessary knowledge, skills, resources, and opportunities within the healthcare system. Effective diabetes care, including assistance for self-management, depends on the availability of trained healthcare professionals; unfortunately, primary healthcare facilities lack sufficient trained HCPs.

## 4.2 Opportunity

Participants frequently discussed and perceived that lack of family support is a major and one of the important barriers for DMSCPs that also affects their motivation to adhere to the required self-care practices. Family members are important sources of physical and emotional support. Family support has also been proven to increase adherence to DMSCPs [40]. A previous study has reported on the significance of family support as an enabler to enhance medication adherence and blood glucose testing in patients with T2DM living in rural areas [41]. This finding implies that family members should be involved in diabetes education programmes; many diabetes care guidelines have also recommended it. Family preferences for food and dietary habits were perceived as obstacles to developing healthy eating habits. This finding is consistent with other studies conducted in different geographic and cultural situations [42].

Financial constraints are perceived as a major barrier to medication adherence, healthy dietary behaviour and regular MBG. Findings are consistent with other similar studies [43]. Our results corroborate previous research that found patients could not perform SMBG as the portable glucometers and glucose strips were not affordable to them [44] In line with previous research, patients with T2DM found it challenging to buy the necessary medications and

healthy foods like fruits and vegetables [15]. HCPs also stated that the absence of essential supplies, including OHAs and glucose strips, along with insufficient access to diagnostic tests such as HbA1c, often impeded their ability to effectively assist patients in managing their DM, and patients with T2DM may feel less inclined to keep frequent follow-ups as a result of all of these deficiencies. The scope of the National Programme for Prevention & Control of Cancer, Diabetes, Cardiovascular Diseases & Stroke (NPCDCS) currently running by government of India public health hospitals could be expanded to cover more effective medications, provide patients with more dosages, include subsidies for glucose monitoring, particularly for HbA1c and inclusion of educational counselling services.

Participants expressed how their busy schedules made it difficult to follow their diets, medication adherence, and exercise, especially for the female participants. These results corroborate studies that showed patients with T2DM had difficulties integrating their regular work schedules with recommendations for exercise, medication adherence, and diet [24]. It is essential to educate patients with diabetes on practical methods to incorporate self-care habits into their work routines, emphasising the advantages of maintaining consistent diabetes self-care practices. In line with findings from previous research, patients' commitment to exercise was hindered by extreme weather conditions [45].

## 4.3 Motivation

Participants need and preference for education related to DMSCPs underline their reflective motivation to do the DMSCPs, which would act as enablers and sustain the SCPs. The participants also emphasised educational material for reference at home as ideal teaching content. Most of the FGD participants proposed combining individual and group education because each has its benefits in terms of learning. Written material in vernacular language, mainly pamphlets, was an excellent way to reinforce knowledge at home with family support. It was also interestingly found that FGD participants proposed to include family members in education. This finding was consistent with the previous study [46]. Participants reported that receiving guidance and support from physicians helped them to perform optimal DMSCPs. Previous research has demonstrated that a positive relationship between a patient and a healthcare professional increases self-care adherence [47].

Interestingly, fear of DM-related symptoms and complications was a motivating factor for practising DMSCPs, according to participants. Previous studies have also highlighted the fear of disease consequences as one of their most typical psychological reactions and facilitators to patients with T2DM SCPs [48]. Perceived self-efficacy (automatic motivation) for adopting an appropriate lifestyle enabled participants to manage DM. This finding is consistent with the earlier results [48, 49]. To enhance motivation for diabetes self-care among individuals with T2DM in India, implementing strategies such as setting small and achievable goals, sharing success stories within the community, and addressing apprehensions through education and support networks are pivotal for fostering a positive outlook and sustained engagement in self-management practices[50].

WHO defined health as "a state of complete physical, mental and social well-being and not merely the absence of disease and infirmity" [51]. However, health policies and programmes currently focus on the physical or biological aspects of disease and illness rather than on diseases' psychological, spiritual, social, and environmental attributes. This is known as the 'biomedical model' of health care [52]. However, NCDs are multifactorial, where individual behaviour is also responsible for their health and illness (bio-psycho-social model).

### 4.4 Strengths and limitations of the study

This study offers insights of patients with T2DM and HCPs on diabetes SCPs for first time from rural areas of India by applying the COM-B model. Second, this study provided the perspective of the key stakeholders, which gives opportunities to address the concerns related to DMSCPs through the collaboration of patients and healthcare providers. Third, FGD included different age groups, gender, and socioeconomic status, and IDI included diverse experienced HCPS increases the findings' transferability. Last but not least, this study was also helpful in other contexts since it raised awareness of the significance of "need assessment" before any health intervention.

Limitations of this study are: first, since the survey was conducted exclusively among patients with T2DM, the findings may not be transferable to patients with type 1 diabetes mellitus. Second, we recruited the study participants from rural areas; thus, the results were limited by geographic location. Despite these limitations, rigorous study methodology enables this to be conceptually transferable.

## 5. Conclusion

The study findings indicate that various factors influence diabetes SCPs from the perspectives of both patients with T2DM and HCPs. For patients with T2DM, limited understanding of DM, beliefs in alternative therapies, concerns about drug side effects, and attitudes towards DM has been acting as barriers to engaging in effective SCPs. The presence of comorbidities and the availability of family support played significant roles in shaping patients' physical and social capabilities to engage in SCPs. Additionally, financial and time constraints, as well as weather conditions, posed challenges to the opportunities for self-care. Physicians who aligned their recommendations with patients' sense of self-efficacy had significant impact on medication adherence. Nonetheless. HCPs faced constraints in delivering patient-centred diabetes care which were limited training opportunities and a lack of essential resources, including medications, diagnostic services, and diabetes educators. A structured diabetes education intervention that aligns with the COM-B model may improve diabetes SCPs and approach should include patient education interventions, enhanced support from HCPs, and the allocation of necessary resources in rural health care setup and training opportunities for rural HCPs.

## Supporting information

**S1 Appendix. Consolidated criteria for reporting qualitative studies (COREQ): 32-item checklist.**
(PDF)

**S2 Appendix. Number of FGDs in different sociodemographic subgroups (total FGDs = 8).**
(DOCX)

**S3 Appendix. Focus group discussion and in-depth interview guide.**
(DOCX)

## Acknowledgments

The authors gratefully acknowledge the FGD participants in the study. They also thank Mrs. Sunita Singh, Public Health Nursing Officer, PGIMER, Chandigarh, Auxiliary Nursing Midwife (ANM), and Accredited Social Health Activists (ASHA) of the selected study site for assisting in the focus-group discussions. We also thank the Department of Health and Family

Welfare, Punjab, for permitting us to conduct the study. The first author was pursuing his Ph. D. through the ICMR's JRF/SRF Scheme.

## Author Contributions

**Conceptualization:** Saurabh Kumar Gupta, P.V.M. Lakshmi, Venkatesan Chakrapani, Manmeet Kaur.

**Data curation:** Saurabh Kumar Gupta, P.V.M. Lakshmi, Manmeet Kaur.

**Formal analysis:** Saurabh Kumar Gupta, P.V.M. Lakshmi, Venkatesan Chakrapani, Ashu Rastogi, Manmeet Kaur.

**Investigation:** P.V.M. Lakshmi, Ashu Rastogi.

**Methodology:** Saurabh Kumar Gupta, P.V.M. Lakshmi, Venkatesan Chakrapani, Manmeet Kaur.

**Project administration:** P.V.M. Lakshmi.

**Resources:** P.V.M. Lakshmi, Ashu Rastogi, Manmeet Kaur.

**Software:** Venkatesan Chakrapani.

**Supervision:** P.V.M. Lakshmi, Ashu Rastogi, Manmeet Kaur.

**Validation:** P.V.M. Lakshmi, Venkatesan Chakrapani, Ashu Rastogi, Manmeet Kaur.

**Writing – original draft:** Saurabh Kumar Gupta.

**Writing – review & editing:** Saurabh Kumar Gupta, P.V.M. Lakshmi, Venkatesan Chakrapani, Ashu Rastogi, Manmeet Kaur.

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
