## [Decision Letter · Decision Letter 0]

4 Oct 2023

PONE-D-23-21411Understanding the diabetes self-care behaviour in rural areas: perspective of patients with type 2 diabetes mellitus and healthcare professionalsPLOS ONE

Dear Dr. Kaur,

Thank you for submitting your manuscript to PLOS ONE. After careful consideration, we feel that it has merit but does not fully meet PLOS ONE’s publication criteria as it currently stands. Therefore, we invite you to submit a revised version of the manuscript that addresses the points raised during the review process.

We look forward to receiving your revised manuscript.

Kind regards,

Steve Zimmerman, PhD

Associate Editor, PLOS ONE

Journal Requirements:

**Additional Editor Comments:**

The manuscript has been evaluated by two reviewers, and their comments are available below.

Reviewer #1 has raised a number of major concerns. The reviewer requests additional details and clarification regarding the methods and analyses, in addition to a reflexivity statement.

Could you please carefully revise the manuscript to address all comments raised?

Reviewers' comments:

Reviewer's Responses to Questions

**Comments to the Author**

1. Is the manuscript technically sound, and do the data support the conclusions?

Reviewer #1: Yes

Reviewer #2: Yes

2. Has the statistical analysis been performed appropriately and rigorously? 

Reviewer #1: N/A

Reviewer #2: I Don't Know

3. Have the authors made all data underlying the findings in their manuscript fully available?

Reviewer #1: Yes

Reviewer #2: Yes

4. Is the manuscript presented in an intelligible fashion and written in standard English?

Reviewer #1: Yes

Reviewer #2: Yes

5. Review Comments to the Author

Reviewer #1: Thank you for the opportunity to read and review this paper. It is an important area of research. Self management behaviours related to diabetes are very poor among the large sections of population in India. Self management behaviours form a major part of the management of a person with diabetes and therefore poor self management behaviours can adversely affect diabetes outcomes. With increasing prevalence of diabetes in India, a detailed understanding of factors influencing diabetes self management behaviours is important. The authors have attempted a good qualitative exploration of the diabetes self management behaviour in Punjab, India. I have some suggestions for improving the paper and hope the authors can work on them to revise their manuscript to make it better.

1. The authors have used the COMB framework of behaviour change for assessing diabetes self management. This is a good approach. The COMB framework has been studied in health related behaviours extensively and provides a solid framework for building the theory of the study. The authors have explained the framework well.

2. In any qualitative research study it would be helpful to clarify the philosophical positions right in the beginning. The authors here seem to be adopting a constructivist approach, but there are strong overtones of positivism in the way the study is conducted as well as presented. There is a mention of removing confounders. This is a positivist idea and not relevant in a social constructivist paradigm. Also the way the results have been presented in tables with percentages is suggestive of a positivist approach. The authors must clarify their philosophical positions in terms of ontology, epistemology and methodology.

3. Sampling and sample selection is a very important aspect of a qualitative study. The authors must explain why they sampled the Fatehgarh Sahib district of Punjab? Was there a rationale for selecting this particular district? They have mentioned that it is a purposive sampling, but not clarified why this district was purposively selected.

4. The authors have provided selection criteria for including persons with diabetes in the FGDs. the rationale for this is not clear. they have said it is maximum variability sampling, however there doesn't seem to be much variability in the sampling method. The sampling strategy must be clarified. It must be borne in mind that while sampling for a qualitative study, more than representativeness of the sample to the whole population, we are interested in getting patients who can provide a wide range of factors that motivate and discourage self management behaviours.

5. The authors must provide some reflexivity on their position as interviewer and note-taken and their influence on the process of interview. Some reflexivity features would include gender of the interviewer, note taken, their experiences with diabetes and diabetes self management, their knowledge and their background. Such details help the reader understand the analysis and interpretation from the perspective of the interviewers.

6. It is mentioned that an uninvolved third party did data analysis. There could be a few advantages in this, but also several disadvantages. knowing what happened during the interviews, and getting a grasp of the emotional cues and environmental cues during the interview will greatly influence the analysis and interpretation. This is not feasible if the person doing the analysis is an uninvolved third party.

7. I have read the results and verbatim quotes. I am not sure if the analysis has been comprehensive. the findings and the quotes associated with them seem to be rather superficial. Most of the discourse is focussed on taking medications. There is very little emphasis on physical activity. Whether this dimension was probed adequately, what factors other than climatic conditions emerged in this? these aspects need to be elaborated. Care of feet, care of footwear, etc. are also conspicuous by their absence. in Rural India, especially among farmers with the practice of barefoot working in the fields, diabetic foot ulcers are extremely common. a detailed exploration of foot care would have been very important. The results have covered aspects of nihilistic attitude towards health and life. this is quite common in the Indian setting, but the exploration of this idea is rather superficial. Is the nihilism associated with depression? is it associated with poverty? is it associated with lack of access to facilities and resources? such exploration could have added more meaning.

Overall, though this is an important study, in its current form I find it lacking in substance and requiring more clarifications in the methods. The manuscript has potential for improvement and can be considered for publication if all the above issues are addressed adequately.

Reviewer #2: Thanks for your valuable paper (Understanding the diabetes self-care behaviour in rural areas: perspective of patients with type 2 diabetes mellitus and healthcare professionals) which enhance the area of qualitative research

Unfortunately, I have no experience in the analysis of qualitative research

So, I accept the current manuscript provided that another professional reviewer in the area of qualitative research gives his/ her opinion

6. PLOS authors have the option to publish the peer review history of their article (what does this mean?). If published, this will include your full peer review and any attached files.

Reviewer #1: **Yes: **Vijayaprasad Gopichandran

Reviewer #2: No

---

## [Author Response · Author response to Decision Letter 0]

22 Dec 2023

We would like to thank the reviewers and editor for their insightful comments and suggestions to improve our manuscript titled “Understanding the diabetes self-care behaviour in rural areas: perspective of patients with type 2 diabetes mellitus and healthcare professionals” (Manuscript ID: PONE-D-23-21411). All of your comments have been incorporated, which has further strengthened the manuscript. The revisions are shown in track changes in the revised copy of the main document. Our responses to the comments are given below.

Note:

The line numbers referred to in the responses below are the line numbers of the revised manuscript with track changes.

Additional Editor Comments:

The manuscript has been evaluated by two reviewers, and their comments are available below.

Reviewer #1 has raised a number of major concerns. The reviewer requests additional details and clarification regarding the methods and analyses, in addition to a reflexivity statement. Could you please carefully revise the manuscript to address all comments raised?

Author response 

Thank you for this opportunity to address the comments from the reviewers. We took care to address all the comments and revised the manuscript.

Reviewer comments to author

Reviewer #1: 

Thank you for the opportunity to read and review this paper. It is an important area of research. Self-management behaviours related to diabetes are very poor among the large sections of the population in India. Self-management behaviours form a major part of the management of a person with diabetes, and therefore, poor self-management behaviours can adversely affect diabetes outcomes. With the increasing prevalence of diabetes in India, a detailed understanding of factors influencing diabetes self-management behaviours is important. The authors have attempted a good qualitative exploration of the diabetes self-management behaviour in Punjab, India. I have some suggestions for improving the paper and hope the authors can work on them to revise their manuscript to make it better.

Author response: We thank you for your encouraging comments and suggestions. We believe considerations of these points significantly enhanced the manuscript. Your guidance has been instrumental in strengthening this crucial aspect of our qualitative research, and we are grateful for the opportunity to address these points.

Our point-by-point responses are provided in the following passages:

Issues to address:

Comment 1: The authors have used the COM-B framework of behaviour change for assessing diabetes self-management. This is a good approach. The COM-B framework has been studied in health-related behaviours extensively and provides a solid framework for building the theory of the study. The authors have explained the framework well.

Author Response: We thank you for your supporting and encouraging words regarding the use of the COM-B model in our manuscript and for showing your interest.

Comment 2: In any qualitative research study, it would be helpful to clarify the philosophical positions right in the beginning. The authors here seem to be adopting a constructivist approach, but there are strong overtones of positivism in the way the study is conducted as well as presented. There is a mention of removing confounders. This is a positivist idea and not relevant in a social constructivist paradigm. Also, the way the results have been presented in tables with percentages is suggestive of a positivist approach. The authors must clarify their philosophical positions in terms of ontology, epistemology and methodology.

Response: 

Thanks for your insightful suggestions regarding our philosophical positions and thoughtful consideration of our qualitative manuscript. In our manuscript, we acknowledge that there may be instances where elements traditionally associated with positivism. Our study sought to identify areas of improvement in their understanding of about diabetes and diabetes self-care and the application of those understandings to enhance diabetes self-care behaviours in rural communities. Our philosophical stance, rooted in constructivism and pragmatism, acknowledges the subjective nature of knowledge formation, considering individual experiences, beliefs, and societal contexts. This perspective allows us to understand the diverse viewpoints of both patients and healthcare providers, particularly regarding diabetes self-care practices in rural areas. (Guba & Lincoln, 1994) By embracing pragmatism, we aim not only to grasp lived experiences but also to derive actionable insights for enhancing diabetes self-care practices among people with T2DM in rural areas. (Denzin & Lincoln, 2005) Balancing qualitative interpretation with a focus on practical application, our approach seeks to comprehensively explore the complexities of Type 2 diabetes self-care behaviours in rural settings. 

References:

Guba, E. G., & Lincoln, Y. S. (1994). Competing paradigms in qualitative research. Handbook of qualitative research, 2(163-194), 105.

Denzin, N. K., & Lincoln, Y. S. (Eds.). (2005). The Sage handbook of qualitative research (3rd ed.). Sage.

(Kindly refer to page no. 24-25; line no. 614-622)

Comment 3: Sampling and sample selection is a very important aspect of a qualitative study. The authors must explain why they sampled the Fatehgarh Sahib district of Punjab? Was there a rationale for selecting this particular district? They have mentioned that it is a purposive sampling, but not clarified why this district was purposively selected.

Response: 

Thank you. The Fatehgarh Sahib district of Punjab was purposively selected as it is showing an increasing prevalence of type-II Diabetes in Punjab.(Anjana RM et al.,2017) Furthermore, this district falls in the field practice area of the Department of Community Medicine and School of Public Health, PGIMER, Chandigarh, and thus we had already built good rapport with the local communities, a desirable feature in a qualitative study for good quality of data.

References: 

Anjana RM, Deepa M, Pradeepa R, Mahanta J, Narain K, Das HK, et al. Prevalence of diabetes and prediabetes in 15 states of India: results from the ICMR–INDIAB population-based cross-sectional study. Lancet Diabetes Endocrinol 2017;5:585–96. https://doi.org/10.1016/S2213-8587(17)30174-2.

(Kindly refer page number:6, line number:166-171)

Comment 4: The authors have provided selection criteria for including persons with diabetes in the FGDs. the rationale for this is not clear. They have said it is maximum variability sampling, however, there doesn't seem to be much variability in the sampling method. The sampling strategy must be clarified. It must be borne in mind that while sampling for a qualitative study, more than representativeness of the sample to the whole population, we are interested in getting patients who can provide a wide range of factors that motivate and discourage self-management behaviours.

Response: We thank you for your query regarding the selection criteria for the FGD participants. 

Rationale: Purposive sampling was employed to deliberately select participants who could offer rich and varied insights into the diabetes self-care practices in rural health care settings. The specific purpose behind this sampling strategy was to capture a diverse range of experiences and perspectives related to diabetes self-care who had diagnosed with T2DM in selected rural area. (Palinkas et al., 2015). Therefore, patients diagnosed with type 2 diabetes mellitus residing in rural areas were selected based on age (>30 years based on NP-NCD programme), socioeconomic status considering the principle of maximum variation of qualitative research which enhances the richness and depth of qualitative data by encompassing a wide array of perspectives and experiences within the chosen population.(Creswell, 2013) 

In revised manuscript, we have also added the identifier along with the supporting quotes. A supplementary table (S2 appendix) has been added to support the information related to the maximum variation of the sample in FGDs.

References:

Palinkas, L. A., Horwitz, S. M., Green, C. A., Wisdom, J. P., Duan, N., & Hoagwood, K. (2015). Purposeful sampling for qualitative data collection and analysis in mixed method implementation research. Administration and Policy in Mental Health and Mental Health Services Research, 42(5), 533-544.

Creswell, J. W. (2013). Qualitative inquiry and research design: Choosing among five approaches. Sage Publications.

(Kindly refer page number: 6 & 12, and line number:179-181& 305-309)

Comment 5: The authors must provide some reflexivity on their position as interviewer and note-taken and their influence on the process of interview. Some reflexivity features would include gender of the interviewer, note taken, their experiences with diabetes and diabetes self-management, their knowledge and their background. Such details help the reader understand the analysis and interpretation from the perspective of the interviewers.

Response: Thanks for asking for more details. We have updated the text in the data collection part of the methodology section.

“One of the manuscript author (male) with a Master's in Public Health (MPH) degree moderated each FGD and IDI, who had more than four years of experience in the development and implementation of non-communicable disease related interventions, and he was assisted by another researcher (female) with an MSc in Nursing qualification as a notetaker who had more than 12 years of experience in field of rural community-based health promotion activities. Before the commencement of the study, the two researchers had received training from experts in qualitative research who had more than 30 years of experience in field of health promotion and qualitative research methodology.” Interviewer’s and notetaker’s background allowed us in combining medical insights with sociological perspectives and thus enriching the study, and offering a more holistic understanding of diabetes self-management within rural contexts with a focus on social structures, cultural influences, and community dynamics. (Smith JA. 2008)

References

Smith, J. A. (Ed.). (2008). Qualitative psychology: A practical guide to research methods. Sage Publications.

(Kindly refer page number: 7& 8 and line number:202-217)

Comment 6: It is mentioned that an uninvolved third party did data analysis. There could be a few advantages in this, but also several disadvantages. knowing what happened during the interviews, and getting a grasp of the emotional cues and environmental cues during the interview will greatly influence the analysis and interpretation. This is not feasible if the person doing the analysis is an uninvolved third party. 

Response: Thank you for your comments on uninvolved third-party data analysis. However, we would like to clarify that the researchers involved in the data collection process analysed the data. The theme and codes were cross-checked by one of the investigator of this study, who was also involved in the monitoring and supervision of the current study. Differences in coding were resolved by discussion and consensus.

(Kindly refer page number: 9 and line number:246-258)

Comment 7: I have read the results and verbatim quotes. I am not sure if the analysis has been comprehensive. the findings and the quotes associated with them seem to be rather superficial. Most of the discourse is focussed on taking medications. There is very little emphasis on physical activity. Whether this dimension was probed adequately, what factors other than climatic conditions emerged in this? these aspects need to be elaborated. Care of feet, care of footwear, etc. are also conspicuous by their absence. in Rural India, especially among farmers with the practice of barefoot working in the fields, diabetic foot ulcers are extremely common. a detailed exploration of foot care would have been very important. The results have covered aspects of nihilistic attitude towards health and life. this is quite common in the Indian setting, but the exploration of this idea is rather superficial. Is the nihilism associated with depression? is it associated with poverty? is it associated with lack of access to facilities and resources? Such exploration could have added more meaning.

Response: 

Point 1: Thank you for your thorough and insightful evaluation of our study results. In our results, we have included information on foot care as well as other self-care activity includes physical activity, diet, blood glucose monitoring. We have highlighted those text and quotes in the results and discussion section (in red colour) with the relevant references, for your kind review.

(Kindly refer the line no. 308 -566 & 586-591, page no. 12-24)

Point 2: We acknowledge that while in our manuscript we did not explicitly use the term "nihilistic attitude,” but we found the negative perceptions towards the diabetes self-care practices among people with T2DM living in rural areas. We have tried to elucidate the explanations behind it with appropriate references. 

“The findings highlight a critical issue surrounding negative perceptions toward diabetes self-care, which might significantly impact people with DM adherence to self-care practices. This aligns with previous research emphasizing the influence of nihilistic attitude on health behaviours and chronic disease management (Hawthorne K, 2001). In the current study the nihilistic attitude toward diabetes self-care may be due to multifaceted factors such as cultural beliefs, socioeconomic challenges, availability of limited and equitable health care resources and psychological distress (diabetes related distress)(Liu Y, 2022). Cultural influences often shape individuals' perceptions of illness and death, impacting their approach to managing chronic conditions (Mendenhall E, 2010). The perception of negative attitudes, particularly in older adults with type 2 diabetes, might originate from co-morbidities and various illnesses, as well as a certainty of end-stage complications (Huang ES, 2007).The fear of succumbing to complications of diabetes pervades individuals with T2DM due to the witnessed impact on family and community (Joshi SR, 2008; Das AK et al., 2022). Additionally, socioeconomic barriers like limited access to healthcare resources or financial constraints might lead to hopelessness, contributing to the belief that self-care efforts are futile (Ghammari F, 2023). Moreover, the psychological burden of living with a chronic condition (multimorbid conditions), inadequate support systems, could foster a fatalistic mindset, impacting people with T2DM outlook on the effectiveness of self-care practices(Gupta SK, 2022: Hagger MS,2017). These interconnected factors might collectively shape the nihilistic perspective toward T2DM self-care management.”

(Kindly refer to page no.: 24-25; line no. 602-622)

References 

Hawthorne, K. (2001). Effect of culturally appropriate health education on glycaemic control and knowledge of diabetes in British Pakistani women with type 2 diabetes mellitus. Health Education Research, 23(6), 227–238.

Mendenhall, E., Seligman, R., Fernandez, A., & Jacobs, E. A. (2010). Speaking through diabetes: Rethinking the significance of lay discourses on diabetes. Medical Anthropology Quarterly, 24(2), 220–239. 

Huang ES, Brown SES, Ewigman BG, Foley EC, Meltzer DO,(2007). Patient perceptions of quality of life with diabetes-related complications and treatments. Diabetes Care, 30, 2478–83. https://doi.org/10.2337/dc07-0499.

Joshi SR, Das A, Vijay V, Mohan V, (2008). Challenges in Diabetes Care in India: Sheer Numbers, Lack of Awareness and Inadequate Control. JAPI,56,443–50

Das AK, Saboo B, Maheshwari A, Nair V M, Banerjee S, C J, et al (2022). Health care delivery model in India with relevance to diabetes care. Heliyon, 8. https://doi.org/10.1016/j.heliyon.2022.e10904.

Ghammari F, Jalilian H, Khodayari‐zarnaq R, Gholizadeh M (2023). Barriers and facilitators to type 2 diabetes management among slum‐dwellers: A systematic review and qualitative meta‐synthesis. Health Sci Rep,6. https://doi.org/10.1002/hsr2.1231.

Gupta SK, Rastogi A, Kaur M, Lakshmi P (2022). Diabetes-related distress and its impact on self-care of diabetes among people with type 2 diabetes mellitus living in a resource-limited setting: A community-based cross-sectional study. Diabetes Res Clin Pract, 191:110070. https://doi.org/10.1016/j.diabres.2022.110070.

Hagger, M. S., Koch, S., Chatzisarantis, N. L., & Orbell, S. (2017). The common sense model of self-regulation: Meta-analysis and test of a process model. Psychological Bulletin, 143(11), 1117–1154

Reviewer#2: 

Thanks for your valuable paper (Understanding the diabetes self-care behaviour in rural areas: perspective of patients with type 2 diabetes mellitus and healthcare professionals) which enhance the area of qualitative research Unfortunately, I have no experience in the analysis of qualitative research So, I accept the current manuscript provided that another professional reviewer in the area of qualitative research gives his/ her opinion

Response: We thank you for your comment on the potential use of the findings to improve the health of people with Diabetes Mellitus. and showing your interest on the manuscript. 

Regards!

---

## [Editor Report · Decision Letter 1]

28 Dec 2023

Understanding the diabetes self-care behaviour in rural areas: perspective of patients with type 2 diabetes mellitus and healthcare professionals

PONE-D-23-21411R1

Dear Dr. Kaur,

We’re pleased to inform you that your manuscript has been judged scientifically suitable for publication and will be formally accepted for publication once it meets all outstanding technical requirements.

Kind regards,

Vijayaprasad Gopichandran

Academic Editor

PLOS ONE
---

## [Editor Report · Acceptance letter]

30 Jan 2024

PONE-D-23-21411R1 

PLOS ONE

Dear Dr. Kaur, 

I'm pleased to inform you that your manuscript has been deemed suitable for publication in PLOS ONE. Congratulations! Your manuscript is now being handed over to our production team.

Kind regards, 

on behalf of

Dr. Vijayaprasad Gopichandran 

Academic Editor

PLOS ONE